# Peptide ELISA and FRET-qPCR Identified a Significantly Higher Prevalence of *Chlamydia suis* in Domestic Pigs Than in Feral Swine from the State of Alabama, USA

**DOI:** 10.3390/pathogens10010011

**Published:** 2020-12-25

**Authors:** Md Monirul Hoque, Folasade Adekanmbi, Subarna Barua, Kh. Shamsur Rahman, Virginia Aida, Brian Anderson, Anil Poudel, Anwar Kalalah, Sara Bolds, Steven Madere, Steven Kitchens, Stuart Price, Vienna Brown, B. Graeme Lockaby, Constantinos S. Kyriakis, Bernhard Kaltenboeck, Chengming Wang

**Affiliations:** 1College of Veterinary Medicine, Auburn University, Auburn, AL 36849, USA; mzh0130@auburn.edu (M.M.H.); fsa0004@auburn.edu (F.A.); szb0116@auburn.edu (S.B.); ksrlipon@gmail.com (K.S.R.); vza0016@auburn.edu (V.A.); poudelanup@gmail.com (A.P.); aak0016@tigermail.auburn (A.K.); srk0002@auburn.edu (S.K.); pricesb@auburn.edu (S.P.); csk0021@auburn.edu (C.S.K.); kaltebe@auburn.edu (B.K.); 2Swine Research and Education Center, Auburn University, Auburn, AL 36830, USA; anderbl@auburn.edu; 3School of Forestry and Wildlife Sciences, Auburn University, Auburn, AL 36849, USA; szb0132@auburn.edu (S.B.); ssm0042@auburn.edu (S.M.); lockabg@auburn.edu (B.G.L.); 4National Feral Swine Damage Management Program, Wildlife Services, Animal and Plant Health Inspection Service, United States Department of Agriculture, Fort Collins, CO 80521, USA; vienna.r.brown@usda.gov

**Keywords:** *Chlamydia suis*, peptide ELISA, PCR, feral swine, USA

## Abstract

*Chlamydia suis* is an important, highly prevalent, and diverse obligate intracellular pathogen infecting pigs. In order to investigate the prevalence and diversity of *C. suis* in the U.S., 276 whole blood samples from feral swine were collected as well as 109 fecal swabs and 60 whole blood samples from domestic pigs. *C. suis*-specific peptide ELISA identified anti-*C. suis* antibodies in 13.0% of the blood of feral swine (26/276) and 80.0% of the domestic pigs (48/60). FRET-qPCR and DNA sequencing found *C. suis* DNA in 99.1% of the fecal swabs (108/109) and 21.7% of the whole blood (13/60) of the domestic pigs, but not in any of the assayed blood samples (0/267) in feral swine. Phylogenetic comparison of partial *C. suis* ompA gene sequences and *C. suis*-specific multilocus sequencing typing (MLST) revealed significant genetic diversity of the *C. suis* identified in this study. Highly genetically diverse *C. suis* strains are prevalent in domestic pigs in the USA. As crowding strongly enhances the frequency and intensity of highly prevalent *Chlamydia* infections in animals, less population density in feral swine than in domestic pigs may explain the significantly lower *C. suis* prevalence in feral swine. A future study is warranted to obtain *C. suis* DNA from feral swine to perform genetic diversity of *C. suis* between commercial and feral pigs.

## 1. Introduction

Obligate intracellular bacteria of genus *Chlamydia* contains 13 recognized species (*C. abortus*, *C. avium*, *C. caviae*, *C. felis*, *C. gallinacea*, *C. muridarum*, *C. pecorum*, *C. pneumoniae*, *C. poikilotermis*, *C. psittaci*, *C. serpentis*, *C. suis*, and *C. trachomatis*) [1,2]. Of all thirteen chlamydial species, only five (*C. suis*, *C. abortus*, *C. pecorum*, *C. psittaci* and *C. trachomatis*) are known to infect pigs [3,4]. The pig is the only known natural host of *C. suis*, and *C. suis* infections in pigs have been reported to be associated with a variety of clinical signs including conjunctivitis, rhinitis, pneumonia, enteritis, and reproductive disorders [4,5,6,7]. However, most recent reports also demonstrated asymptomatic *C. suis* infections in pigs in Austria, Belgium, China, Germany, Japan, Italy, and Switzerland [7,8,9,10,11].

Feral swine are considered to be the single most invasive animal species in the United States and have expanded from 17 to 38 states in the last 30 years [12]. In the state of Alabama, the home range of feral swine has also spread to all 67 counties [13]. It has been reported that there is a high probability of interaction between domestic pigs having outdoor access and feral swine in certain geographic regions. Close contact between domestic pigs and feral swine is considered a risk factor for transmission of pathogens, including *C. suis* [14]. Wahdan et al. reported a low prevalence of *Chlamydia* DNA in the investigated feral swine populations (1.4%, 4/292) in Switzerland. In addition, microimmunofluorescence test was performed to test for antibodies to *Chlamydia* spp. in sera from hunter-killed feral swine harvested during the 2006–2009 hunting seasons in three Italian regions, and 63.6% (110/173) tested sera were shown to have antibody titers to chlamydiae >1:32 [15].

While *C. suis* was isolated from domestic pigs in the USA [5,16], little is known about the prevalence and diversity of *C. suis* in pigs in the USA. Therefore, the present study was undertaken to investigate the molecular and serological prevalence of *C. suis* in feral and domestic pigs in Alabama. In addition, phylogenetic analysis using both *ompA* and a *C. suis*-specific MLST typing scheme was performed to analyze the *C. suis* diversity.

## 2. Results

*C. suis*-specific peptide ELISA determined a significantly higher prevalence of anti-*C. suis* antibodies in domestic pigs than in the feral swine. The anti-*C. suis* antibody was detected in 13.0% (240/276) of whole blood samples in the feral swine, being significantly lower than 80.0% positivity (12/60) in domestic pigs (Figure 1A) (*p* < 10^−4^). In addition, the percentage of the whole blood samples with a strong positive antibody level (OD value > 1.0) was significantly higher in domestic pigs than in feral swine (46.7%, 28/60 vs. 2.2%, 6/276; *p* < 10^−4^) (Figure 1A).

DNA sequencing following FRET-qPCR determined that only *C. suis*, no other chlamydial species, was identified in swine samples of this study. In a similar trend as indicated by peptide-ELISA, *C. suis* DNA was found in 21.7% (13/60) of the whole blood of domestic pigs, but not in any of the whole blood samples from feral swine (0/276) (Figure 1B). While the fecal swabs were not available from feral swine in this study, *C. suis* DNA was identified in 99.1% (108/109) of fecal samples of domestic pigs in this study. Paired blood and fecal samples were collected from 60 of 109 domestic pigs in this study, and all 13 pigs which were found to be *C. suis* positive in whole blood were also positive in their fecal swabs (Figure 1B).

Seven distinct partial *ompA* sequences encompassing the variable domain 1 and 2 (VD1-2) from 24 pig *C. suis* isolates were identified in this study (Figure 2). Compared with the existing *ompA* sequences deposited in GenBank, the partial *ompA* VD1-2 sequences from *C. suis* strains identified in this study are highly polymorphic. Still, these seven highly polymorphic *ompA* sequences identified in this study cluster and differ from those of other isolates of other countries, but showed a high similarity with a swine isolate in Germany (AY687634) (Figure 2). While considerable *ompA* sequence variation was observed in this study, three identical sequences (MT997040, MT997041, and MT997042) were found from fecal swabs of three domestic pigs (Figure 2).

Bayesian phylogenetic analysis of concatenated nucleotide sequences of seven MLST was performed on 11 *C. suis* sequences identified in this study, and these sequences are compared with 17 other *C. suis* sequences deposited in GenBank from six countries (Germany, Switzerland, Italy, USA, Japan, and China). The *C. suis* isolates identified in this study were found to be grouped in separate but diverse sub-clades (Figure 3).

## 3. Discussions

In this study, *C. suis*-specific peptide ELISA and highly specific and sensitive FRET-qPCR were performed to investigate *C. suis* prevalence in domestic and feral pigs in Alabama. Both peptide ELISA and FRET-qPCR indicated a high prevalence of *C. suis* in domestic pigs. While fecal swabs from feral pigs were not available for FRET-qPCR in this study, peptide ELISA showed a positive but much lower anti-*C. suis* antibody in feral swine than in domestic pigs (13.0% vs. 80.0%, respectively; *p* < 10^−4^; Figure 1A).

In a calf model to explore the prevalence of natural *Chlamydia* species, Jee et al. reported that the group size of calves correlated positively with chlamydial infection in quadratic regression, and a doubling of the group size was associated with a four-fold increase in frequency and intensity of *Chlamydia* infection [17]. This might explain well the significantly lower *C. suis* prevalence in feral swine than in domestic pigs as the population density of feral swine is much lower than that of domestic commercial pigs. The observation of this study further verifies the notion that crowding strongly enhances the frequency and intensity of highly prevalent *Chlamydia* infections in animals [17].

*C. suis OmpA* PCR and the *C. suis* specific MLST scheme demonstrated that the highly prevalent *C. suis* in domestic pigs in this study are also highly polymorphic as reported elsewhere worldwide [7,11,18]. Phylogenetic analyses showed that the *ompA* VD1-2 gene fragment of the *C. suis* strains in this study is highly polymorphic. *C. suis* MLST analysis also suggested a shared ancestry of *C. suis* strains in the USA with those described in Europe.

*C. suis* is often found in the intestine [19,20], conjunctiva [21], the genital tract [22], nasal swabs [23], lung tissue [24], and the liver of aborted fetuses [25]. In this study, the prevalence of *C. suis* DNA in fecal swabs was significantly higher when compared to those taken from whole blood samples (Figure 1). This result is most likely due to the gastrointestinal tract being the primary site of infection and chlamydial replication. This finding agrees with the report by Li et al., showing 8.0% positivity of *C. suis* in whole blood and 60.0% positivity in feces [11]. In addition, anti-*C. suis* antibody prevalence was 80% the blood of the assayed domestic pig in this study while *C. suis* DNA was present in 99.1% of the fecal samples (Figure 1). The difference in serological and molecular prevalences might be due to the limited sensitivity of the peptide ELISA used in this study, and the antibody response might be too weak to be detected in the early stage of *C. suis* infection.

The microimmunofluorescence (MIF) test is the standard serological assay for species-specific detection of antibodies against chlamydiae [26], but shows cross-reactivity and poor sensitivity [27,28,29]. Rahman et al. established a species-specific molecular serology for different chlamydial species based on the defined species-specific immunodominant B cell epitopes [30,31]. In the present study, this previously validated *C. suis*-specific peptide ELISA was used to detect antibodies in feral and domestic pigs. We reported that 13% of the assayed feral swine were positive for *C. suis* antibodies which is lower than 63.6% positivity in feral swine in Italy [15]. Specificity of MIF and peptide ELISA may explain, in part, the different positivity of *C. suis* antibodies in Italian study and this work.

Samples of small populations of domestic pigs and wild feral swine from the state of Alabama were available in this study. Future study is warranted to collect more samples from different regions of the USA. and investigate the overall prevalence of *C. suis* in the entire USA. In addition, fecal samples from feral swine in USA should be obtained to compare genetic diversities of *C. suis* between commercial and feral pigs

In conclusion, the serological and molecular surveys in this study indicate that *C. suis* infection in domestic pigs is common while a significantly lower *C. suis* prevalence is found in feral swine. Molecular typing of detected strains suggests that *C. suis* in the USA. are genetically diverse as the global diversity of this pathogen reported in other countries.

## 4. Material and Methods

### 4.1. Ethics Statement

Protocols for the collection of swine samples in this study were reviewed and approved by the Auburn University Institutional Animal Care and Use Committee (Approval number: 2017-3143).

### 4.2. Collection of Whole Blood Samples from Feral Swine

Between July 2019 and March 2020, feral swine (n = 276) were trapped at a 4515-hectare privately-owned land in Bullock County in Alabama as described [32]. The property lies within the Upper Coastal Plain physiographic region. It was estimated that the wild pig density on this property is 15.5 pigs/km^2^ which is greater than the average density of 6–8 pigs/km^2^ in the region [33]. The feral swine were captured using the Jager Pro Hog Control Systems corral trap with a remotely activated gate. The gate was be activated via a cellular network to close when the feral swine were seen on camera inside the trap, and a small caliber rifle was used to euthanize the feral swine.

Whole blood samples (n = 276) were collected into 5 mL EDTA tubes and were transported on ice to the research lab within three hours of sample collection.

### 4.3. Collection of Whole Blood and Fecal Swab Samples from Domestic Pigs

Between July and August of 2020, 60 EDTA whole blood samples and 109 fecal swab samples were collected from 109 domestic pigs at the Auburn University Swine Research and Education Center (AUSREC). AUSREC is a breed-to-finish swine production facility providing education and research to students and quality pork products to the community. Whole blood samples were collected into 10 mL EDTA tubes and were transported on ice to the research lab within three hours of sample collection. Fecal swabs were collected into sterile Eppendorf tubes containing 400 μL 1× phosphate buffer solution, and were transported to the research lab within 3 h of sample collection.

### 4.4. Peptide ELISA to Detect Anti-C. suis Antibodies in the Plasma Samples

The collected EDTA whole blood samples from feral swine and domestic pigs were centrifuged at 1000× *g* for 10 min, and 200 µL plasma was transferred to microcentrifuge tube and stored at −20 ℃ for peptide ELISA. The remaining blood samples were transferred to microcentrifuge tubes and stored in −80 ℃ until nucleic acid extraction and PCR were performed as described below.

The *C. suis* species-specific peptide antigens as well as the protocol of running peptide ELISA were used as previously validated and described [30,34]. *C. suis* peptide antigens were chemically synthesized with N-terminal biotin followed by a serine-glycine-serine-glycine spacer mixture Thermo Fisher Scientific, Waltham, MA, USA). The peptide mixture consisted of 30 peptide antigens with equal molar amount, and coated on streptavidin-coated white microtiter plates (Fisher Scientific, Roskilde, Denmark).

The plasma was tested for anti-*C. suis* IgG with a horseradish peroxidase (HRP)-conjugated goat anti-pig IgG (h + l) cross-adsorbed antibody (Bethyl Laboratories, Inc., Montgomery, TX, USA) by colorimetric ELISA [30]. Using the titration of diluted sera and conjugates, the optimal concentrations of sera (1:40 dilution) and conjugates (1:80,000 dilution for the polyclonal IgG-HRP conjugate) were determined. Background of each serum was determined by the wells coated with DMSO. Plasma samples were run in both peptide and DMSO control in a replicate. Optical density was measured at 450  nm (Tecan Spectrafluor Plus reader, Madison, WI, USA). The OD values for individual plasma sample was calculated after background correction with 110% subtraction from the average of each raw sample signal. The samples with an OD value below 0.19 were considered negative, positive when the OD value was above 0.2, and strong positive as OD > 1.0.

### 4.5. Extraction of Nucleic Acids from Whole Blood Samples and Fecal Swabs

The High-Pure PCR Template Preparation Kit (Roche Diagnostics, Indianapolis, IN USA) was used to extract total nucleic acids from whole blood according to the manufacturer’s instructions and described previously [11,32]. In brief, whole blood (400 µL) was mixed with equal volume of binding buffer followed by homogenization and digestion with proteinase K (10% of total volume). Nucleic acid was eluted in the final volume of 200 µL. DNA extraction from fecal samples followed the same procedure as described above [11]. Each fecal swab sample was mixed with 400 µL of binding buffer, and eluted to the final volume of 100 µL.

### 4.6. Chlamydia FRET-qPCR

The *Chlamydia* FRET-PCR used in this study followed the protocols described by [35,36]. In brief, 10 µL of the extracted DNA was added to a 10 µL reaction mixture containing 5× PCR FRET buffer, 400 µM dNTP (Roche Diagnostics GmbH, Indianapolis, IN, USA), 0.34 units of Platinum Taq DNA Polymerase (Invitrogen), 1 µM of each forward and reverse primer (Integrated DNA Technologies, Coralville, Iowa, USA) and a final volume of Molecular grade Nuclease-free water. This PCR amplified a 168-bp fragment of the *Chlamydia* spp. 23S rRNA gene, and was able to detect all 11 *Chlamydia* species with a detection sensitivity of single copy/reaction. PCR amplification was performed in a LightCycler 480-II real-time PCR platform using a high-stringency 18-cycle step-down temperature protocol: 6 × 10 s, 95 ℃; 10 s, 64 ℃; 10 s, 72 ℃; 9 × 10 s, 95 ℃; 10 s, 62 ℃; 10 s, 72 ℃; 3 × 10 s, 95 ℃; 10 s, 60 ℃; 10 s, 72 ℃; followed by 30 low-stringency cycles: 30 × 10 s, 95 ℃; 10 s, 56 ℃; 10 s, 72 ℃. The PCR products were further verified by electrophoresis followed by DNA sequencing (ELIM Biopharmaceuticals, Hayward, CA, USA) using both primers.

### 4.7. C. suis-Specific ompA-PCR

For the investigation of the polymorphisms in the *C. suis ompA* gene, a set of previously validated primers [11] were used in this study to amplify the *ompA* VD 1-2 (amplicon size: 491 bp) in 24 *C. suis*-positive samples (10 whole blood and 14 fecal swabs) from domestic pigs. PCR amplification was performed with SYBR system in a LightCycler 480-II real-time PCR platform using a high-stringency 18-cycle step-down temperature protocol as mentioned above. The PCR products were further verified by electrophoresis followed by DNA sequencing (ELIM Biopharmaceuticals, Hayward, CA, USA) using both primers, and ompA sequences were submitted to the GenBank (Accession numbers: MT997036 to MT997042).

### 4.8. C. suis-Specific MLST PCRs

In this study, a *C. suis*-specific MLST typing scheme based on a previously published Chlamydiales MLST scheme [37,38] was performed to amplify seven *C. suis* housekeeping genes in *C. suis* positive samples. These housekeeping genes were selected using the criteria that they are widely separated on the chromosome and not adjacent to a putative outer membrane, secreted, or hypothetical proteins that might be under diversifying selection while it is assured that each locus had a similar extent of nucleotide substitutions to ensure consistency [39]. PCR amplification of the seven housekeeping genes was performed with SYBR system in a LightCycler 480-II real-time PCR platform following the above-mentioned protocol.

The products of all seven *C. suis* MLST PCRs were sent to ELIM Biopharmaceuticals (Hayward, CA, USA) for DNA sequencing using both primers, and the GenBank accession numbers were obtained (Table 1).

### 4.9. Phylogenetic Analysis

Phylogenetic analyses were performed using the 489 bp variable region of the *ompA* gene and the concatenated *C. suis* MLST sequences. For *ompA*, a total of 22 sequences consisting of 7 from this study and 15 publicly available sequences obtained from GenBank were aligned using the ClustalX 1.83. A Bayesian phylogenetic tree was created using an alignment of a total of 11 concatenated MLST sequences from this study, and 14 additional strains from Switzerland, USA, Italy, China, and Austria. Based on these alignments, phylogenetic trees were constructed by the neighbor-joining method using the Kimura 2-parameter model with MEGA 6.0. Bootstrap values were calculated using 500 replicates.

### 4.10. Statistical Analysis

All statistical analyses were performed with the Statistica 7.0 software package (StatSoft, Inc., Tulsa, OK, USA). Chi-squared test was performed to compare the positivity of *C. suis* DNA and antibody between domestic pigs and feral swine, and between blood samples and fecal swabs in domestic pigs. Difference at *p* ≤ 0.05 was considered significant.

## Figures and Tables

**Figure 1 pathogens-10-00011-f001:**
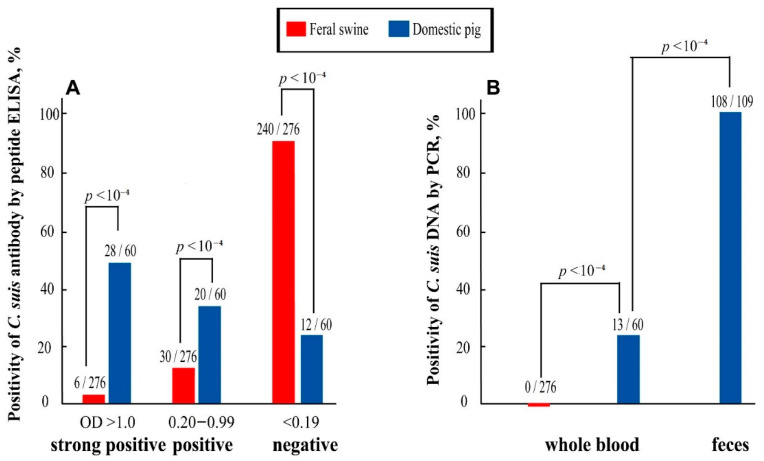
Significant higher prevalence of *C. suis* in commercial pigs than in the feral swine determined by PCR and peptide ELISA. (**A**) Species-specific peptide-ELISA determined a significantly higher prevalence of *C. suis* antibodies in commercial pigs than in feral swine. (**B**). FRET-qPCR and DNA sequencing identified *C. suis* DNA in 21.7% (16/30) of the whole blood and 99.1% of feces (108/109) of commercial pigs, but not in the whole blood of feral swine (0/276).

**Figure 2 pathogens-10-00011-f002:**
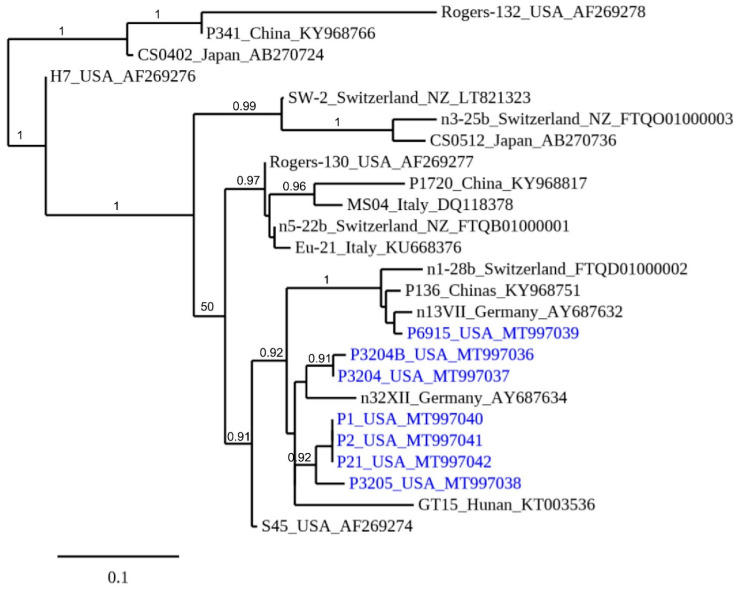
Phylogenetic tree of the *ompA* variable domains 1-2. A 489-bp fragment encompassing *C. suis* ompA VD1-2 of seven porcine *C. suis* strains identified in this study (in blue font; name of strain, country, accession number) are compared with 18 other *C. suis* sequences deposited in GenBank from six countries (Germany, Switzerland, Italy, USA, Japan, and China). Branch lengths are measured in nucleotide substitutions and numbers show branching percentages in bootstrap replicates. Scale bar represents the percent sequence diversity.

**Figure 3 pathogens-10-00011-f003:**
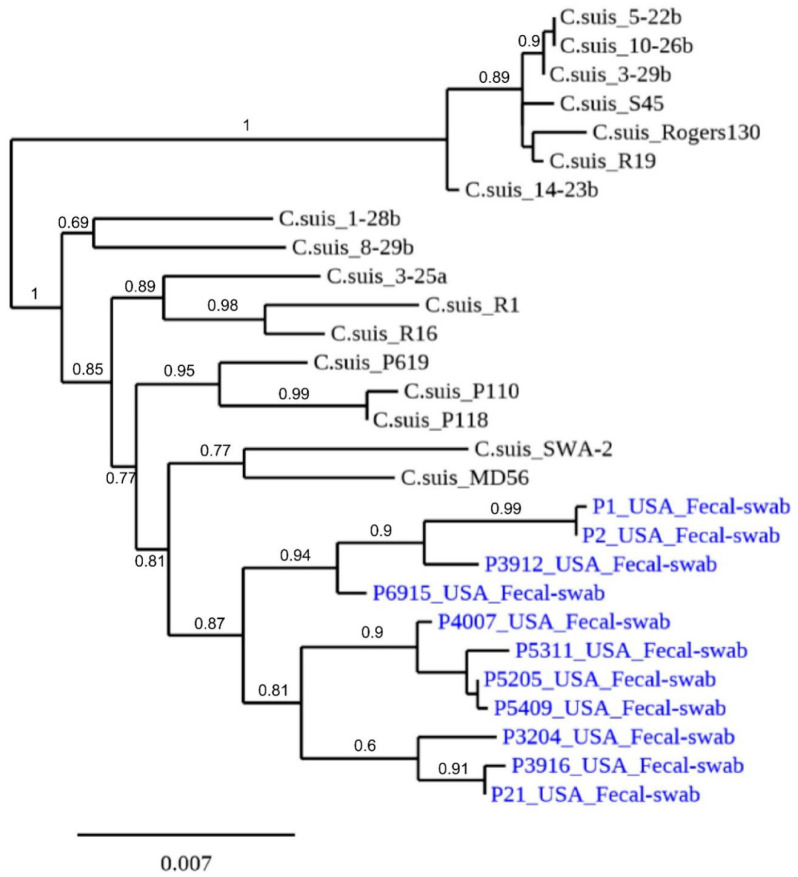
Bayesian phylogenetic analysis of the concatenated sequences of seven MLST fragments of *C. suis* strains. The concatenated nucleotide sequences of seven MLST of 11 *C. suis* identified in this study (in blue font; name of strain, country) are compared with 17 other *C. suis* sequences deposited in GenBank from six countries (Germany, Switzerland, Italy, USA, Japan, and China). Branch lengths are measured in nucleotide substitutions and numbers show branching percentages in bootstrap replicates. Scale bar represents the percent sequence diversity.

**Table 1 pathogens-10-00011-t001:** GenBank accession numbers for seven MLST genes of eleven *C. suis* isolates in this study.

Sample ID	*gatA*	*oppA*	*hflx*	*gidA*	*enoA*	*hemN*	*fumC*
**FS-3204**	MW240765	MW240776	MW240787	MW240798	MW240809	MW240820	MW240831
**FS-3912**	MW240766	MW240777	MW240788	MW240799	MW240810	MW240821	MW240832
**FS-3916**	MW240767	MW240778	MW240789	MW240800	MW240811	MW240822	MW240833
**FS-4007**	MW240768	MW240779	MW240790	MW240801	MW240812	MW240823	MW240834
**FS-5205**	MW240769	MW240780	MW240791	MW240802	MW240813	MW240824	MW240835
**FS-5311**	MW240770	MW240781	MW240792	MW240803	MW240814	MW240825	MW240836
**FS-5409**	MW240771	MW240782	MW240793	MW240804	MW240815	MW240826	MW240837
**FS-6915**	MW240772	MW240783	MW240794	MW240805	MW240816	MW240827	MW240838
**FS-1**	MW240773	MW240784	MW240795	MW240806	MW240817	MW240828	MW240839
**FS-2**	MW240774	MW240785	MW240796	MW240807	MW240818	MW240829	MW240840
**FS-21**	MW240775	MW240786	MW240797	MW240808	MW240819	MW240830	MW240841

## Data Availability

The data presented in this study are openly available, and nucleotide sequences are submitted in the GenBank with accession numbers.

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
