# Peer review of "Peptide ELISA and FRET-qPCR Identified a Significantly Higher Prevalence of Chlamydia suis in Domestic Pigs Than in Feral Swine from the State of Alabama, USA"

_pathogens, 2020, doi:10.3390/pathogens10010011_

Round 1
Reviewer 1 Report
The manuscript describes results of interesting study focusing on Chlamydia suis – microorganism which impact on health and productivity of pigs is not fully recognized. The main aim of the study was to investigate occurrence/prevalence of C. suis in domestic and feral pigs in USA. The study design was appropriate and manuscript is written correctly. Contents of all sections are adequate. Although I mode some comments pointing several flaws.
Major comments:
- The title of article suggests that describes prevalence of suis in USA. In my opinion targeted populations of feral boars as well as domestic pigs were too small and concentrated to justify use of term prevalence in context of whole USA.
- Why fecal swabs were not collected form feral boars? Lack of results from these samples is one the serious weakness of the study.
- In discussion section disparity between antibody prevalence (80%) and presence of suis in fecal samples (99,1%) should be explained.
Minor comments
- Line 33: “13recognized” – space is missing
- Lines 35-36: What about trachomatis?
- Lines 58-62: Starting results description from negative results in my opinion, makes reading more difficult.
- Figure 1. Legend should be centered because refers to both A and B chart.
- Lines 156-157: Why did you collect different numbers of fecal swabs and whole blood samples?
- Line 165: For ELISA you have used plasma but previous descriptions of the method refers to serum samples. Do you think it may affect method and its results?
- In-text citations of references should be unified, for example lines 147, 176, 187.
Author Response
We greatly appreciate the constructive comments from this reviewer, followed these excellent comments carefully, and made revisions accordingly! We admit the limitations in this study, and modify the manuscript title and provide the related discussions. Thank you!
The manuscript describes results of interesting study focusing on Chlamydia suis – microorganism which impact on health and productivity of pigs is not fully recognized. The main aim of the study was to investigate occurrence/prevalence of C. suis in domestic and feral pigs in USA. The study design was appropriate and manuscript is written correctly. Contents of all sections are adequate. Although I mode some comments pointing several flaws.
Major comments:
- The title of article suggests that describes prevalence ofsuis in USA. In my opinion targeted populations of feral boars as well as domestic pigs were too small and concentrated to justify use of term prevalence in context of whole USA.
Response: We agree with the reviewer that the populations of domestic pigs and feral boars in this study were only from Alabama and small. The feral boars used is this study was convenient samples from a previous investigation (Poudel A, Hoque MM, Madere S, Bolds S, Price S, Barua S, Adekanmbi F, Kalalah A, Kitchens S, Brown V, Wang C, Lockaby BG. Molecular and Serological Prevalence of Leptospira spp. in Feral Pigs (Sus scrofa) and their Habitats in Alabama, USA. Pathogens. 2020;9(10):857.), and we were not able to obtain more samples. We keep in mind the advice from the reviewer, and will collect more samples from different regions. In responding the question from the reviewer, we modified the title (indicated that the samples were only from Alabama), and provided the limitations of this study in Discussion (lines 142-146).
Why fecal swabs were not collected form feral boars? Lack of results from these samples is one the serious weakness of the study.
Response: The collection of fecal samples in feral boars were for the detection of pathogenic Leptospira while the fecal samples were not collected. We used convenient samples from the leptospiral project for this study. Yes, we agree with the reviewer that the lack of fecal samples from feral boars in this study is one serious weakness of this study. We will try to get funding and collect those samples from more feral boars in the future. Also, we provided a Discussion related to this weakness in the manuscript (Lines 142-146).
In discussion section disparity between antibody prevalence (80%) and presence of suisin fecal samples (99,1%) should be explained.
Response: The discussion is provided in following the reviewer’s advice (Lines 129-133).
Minor comments
Line 33: “13recognized” – space is missing
Response: the typo is corrected. Thank you!
Lines 35-36: What about trachomatis?
Response: we followed the reviewer’s advice and add C. trachomatis here (Lines 36-37).
Lines 58-62: Starting results description from negative results in my opinion, makes reading more difficult.
Response: We fully agree with the reviewer, and modified the sentence accordingly: showing the positive results other than the negative results (Lines 59-60).
Figure 1. Legend should be centered because refers to both A and B chart.
Response: We moved the legend in the center of the figure as suggested by the reviewer.
Lines 156-157: Why did you collect different numbers of fecal swabs and whole blood samples?
Response: Thanks for the sharp eyes from the reviewer about the different numbers of fecal swabs and whole blood samples in the domestic pigs. In following the Auburn University Swine Research & Education Center (AUSREC), we did not get permission to collect whole blood samples from all investigated pigs. The management at AUSREC may want to reduce the stress to the certain groups of pigs, and in these groups, only fecal swabs were collected.
Line 165: For ELISA you have used plasma but previous descriptions of the method refers to serum samples. Do you think it may affect method and its results?
Response: We personally think that the ELISA results from plasma and sera should be similar, but we did not perform an experiment to support this statement. For the convenience of this study, the whole blood samples were collected into EDTA tubes so that we will have plasma samples for ELISA, and whole blood for DNA extraction/PCR/DNA sequencing. To obtain sera samples, we would have to collect two tubes of blood samples from each pig.
In-text citations of references should be unified, for example lines 147, 176, 187.
Response: I apologize for the silly mistakes here. The citations of the references are now unified. Thanks for the reviewer’s sharp eyes.
Reviewer 2 Report
The paper write by Monirul Hoque and coworkers treated an intresting topic.
C. Suis prevalence and DNA sequencing revealed significant genetic diversity of the C. suis identified in this study. Material and Methods are well described as The results. However, the discussion could be improved to better outline the importance of the results obtained. In addition, there are some typos e.g. line 33
these should be correct.
Author Response
The paper write by Monirul Hoque and coworkers treated an intresting topic. C. suis prevalence and DNA sequencing revealed significant genetic diversity of the C. suis identified in this study. Material and Methods are well described as The results. However, the discussion could be improved to better outline the importance of the results obtained. In addition, there are some typos e.g. line 33 these should be corrected.
Response:
1) Line 33. The typo was corrected.
2) We followed the reviewer’s advice and provided more discussions regarding the importance of the results in this study (Lines 129-133; Lines 142-146).
Round 2
Reviewer 1 Report
Thank you for all corrections and explanations. I have no further comments to the manuscript.